# Comparison of Dynamic Vibration Control Techniques by the GFRPU and/or LTMD System

**DOI:** 10.3390/polym14224979

**Published:** 2022-11-17

**Authors:** Jae-Hyoung An, Jun-Hyeok Song, Hye-Sook Jang, Pil-Sung Roh, Hee-Chang Eun

**Affiliations:** 1Department of Architectural Engineering, Kangwon National University, Chuncheon 24341, Republic of Korea; 2R&J Structural Engineering Consultants, Wonju-si 26392, Republic of Korea

**Keywords:** structural reinforcement, lever-type tuned mass damper, glass fiber-reinforced polyurea, constraint, control force, ductility

## Abstract

Reinforced concrete (RC) structures with non-seismic reinforcement details are vulnerable to earthquakes. This experimental study evaluates the efficiency of three techniques to alleviate the dynamic responses of existing structures: glass fiber-reinforced polyurea (GFRPU) reinforcement, a lever-typed tuned mass damper (LTMD) system, and a hybrid system of GFRPU and LTMD reinforcements. The lateral-resisting capacity and ductility of the GFRPU reinforcement specimen were enhanced by the material characteristics, and the dynamic responses were alleviated. The LTMD control specimen controlled the dynamic responses by the passive control system of the tuned mass damper (TMD), and the control forces to sustain its geometric motion were exerted on the specimen. The hybrid system was designed to control the dynamic responses by the GFRPU reinforcement and the LTMD control system. Four specimens, including an unreinforced specimen, were tested under external excitations, including the El Centro earthquake. The vibrations were more controlled in the order of the GFRPU reinforcement specimen, the LTMD control specimen, and the hybrid control specimen. The hybrid system was evaluated as excellent for seismic reinforcement, such as preventing abrupt failure with the lateral-resisting capacity and ductility of GFRPU and improving the dynamic control capacity by LTMD.

## 1. Introduction

Due to the recent earthquakes in Korea, social interest in the seismic reinforcement of existing structures has emerged. In addition to seismic reinforcement design to enhance the lateral rigidity by braces, shear walls, etc., in structures, dynamic control systems have been considered. The systems control the vibration responses by dissipating the kinetic energy or activating control forces.

Many control systems have been developed to alleviate dynamic responses, and one of these is a tuned mass damper (TMD) passive control system. It transfers part of the vibration energy of the structure to the TMD by tuning with the primary frequency of the structure and dissipating the energy by inertial force [1,2,3,4,5,6]. A lever-type tuned mass damper (LTMD) [7] expanded the TMD design concept and used the action of the control forces required for constraining the inter-story displacement responses. Verification through empirical experiments for vibration control, such as the optimal properties of LTMD, has been required.

A TMD system can control the vibration response of the structure, even though the weight is much smaller than that of the structure. The TMD control system can prevent the deterioration of a residential environment and damage or complete destruction of structures due to external excitations such as earthquakes and wind. When the displacement responses of the structure subjected to external excitations are constrained, the control forces required for satisfying the conditions act on the structure. The LTMD is a control system that utilizes the TMD concept and the control force to constrain the dynamic responses of the lever. The displacement responses at the upper and lower ends of the lever, $y_{T}$ and $y_{b}$, are restricted by a hinge installed at an arbitrary position in the lever direction (Figure 1). The dynamic responses are described by the rotation θ. It was reported in a previous study [7] that the vibration responses were more effective than those of the TMD.

Polyurea is a material produced by a chemical reaction between an isocyanate-based polymer and an amine-based curing agent. Coating polyurea has been utilized for enhancing the blast and impact resistance capacity of structural members [8,9,10,11,12,13] because of its excellent quasi-static mechanical properties and dynamic mechanical properties. Alizadeh and Amirkhizi [14] reported that milled carbon fiber polyuria strengthened the polyuria with milled carbon fibers and controlled the temperature rise due to dissipated heating. Anas et al. [15] reviewed major publications related to the blast response of the RC slabs, including failure modes such as the spalling of concrete and concrete scabbing. Lee et al. [16] developed stiff-type polyuria by varying the prepolymer/hardener ratio of flexible-type polyuria as a structural retrofitting and strengthening material. There have been studies on the strengthening method of discrete fiber-reinforced polyurea (DFRP), mixing polyurea with high tensile strength and elongation and chopped glass fibers for further improving the tensile strength. The polyurea and chopped glass fibers are simultaneously sprayed through two separate guns. It was reported [17,18,19] that the DFRP reinforcement enhanced the flexural and shear capacities, as well as the ductility.

Glass fiber-reinforced polyurea (GFRPU) is a composite elastomer whose tensile strength is further improved by mixing milled glass fibers with polyurea of high tensile strength and elongation. It is easy to construct by spraying on the member surface, and it has the advantage of a low construction cost. The GFRPU coating technique sprays through a single nozzle, unlike the DFRP method. The GFRPU reinforcement improves the lateral rigidity and ductility of the reinforced concrete (RC) members by the material characteristics. The structural performance of the RC members coated with GFRPU has been investigated, and the validity was evaluated by several researchers [20,21,22].

Four specimens, including an unreinforced specimen, GFRPU, LTMD, and hybrid reinforcement specimens, were produced and tested under external excitations, including the El Centro earthquake. The intact specimens were designed with non-seismic reinforcement details. The GFRPU reinforcement technique had the merit of improving the ductility as well as the lateral rigidity of structural members. The LTMD control system was effective in controlling the dynamic responses of the entire structure. The hybrid control system was designed to control the vibration of entire structures by the LTMD and secure the lateral rigidity and ductility of the individual members by the GFRPU. This study investigates the effectiveness and control capacity of the GFRPU reinforcement technique and the LTMD control system. It was observed that the displacement responses were more controlled in the order of the GFRPU reinforcement, the LTMD system, and the hybrid system.

## 2. Research Significance

The motivation for conducting this study was to prove the reinforcement method using material characteristics of GFRPU and the LTMD control system for reducing seismic responses. It compared the control effect by the GFRPU reinforcement and/or the LTMD control system. It has been verified through experiments that the dynamic responses were controlled by these reinforcing materials and systems. It was observed that the reinforcement effect was enhanced in the order of the GFRPU reinforcement, the LTMD control system, and the hybrid system of GFRPU reinforcement and the LTMD system.

## 3. Specimens

A structure can be converted into a single degree of freedom (SDOF) model by assuming its primary vibration mode. The specimens were described as an SDOF model to be composed of a column of a lateral resistance member and an upper slab corresponding to a mass. The column in the specimens was modeled as the lateral resistance members of the structure, and the slab corresponded to the mass supporting the dead load in the direction of gravity. The TMD was designed using the properties of the SDOF system model proposed by Den Hartog [1]. It was installed at the position to represent the largest mode value in the primary vibration mode. Although the LTMD was a two-DOF dynamic system at the upper and lower ends of the lever, it was designed in the same way as the TMD of an SDOF system because the responses at both ends can be described by the rotation angle of the lever, θ, as shown in Figure 1.

Four specimens, including an unreinforced specimen, were prepared to verify the displacement control effects by the GFRPU reinforcement, the LTMD system, and the hybrid system (Figure 2). The dynamic responses of these specimens were evaluated and compared.

### 3.1. Materials

Polyurea (Figure 3) is an elastomer with excellent constructability, which hardens within 30 s at room temperature, having remarkable properties, such as tensile strength, tear, impact, and abrasion resistance, and durability and restoring force. It has been widely used as a waterproofing material because of its fast curing time and is relatively insensitive to moisture.

The higher tensile strength of polyurea can be ensured by adding milled glass fiber, as shown in Figure 4, with high tensile strength. The GFRPU becomes a composite elastomer with a high tensile strength along with the intrinsic ductility properties of polyurea. It can be utilized to strengthen the structural members. A careful manufacturing process was required to prevent the milled glass fibers from being separated from the polyurea and agglomerating during mixing so that they could be sprayed through a nozzle. It was found through repeated experiments that the 300 μm milled glass fiber was prevented from being separated at a centrifugal speed of 500 rpm. The fabricated GFRPU was smoothly applied to the surface of the specimens through a nozzle and a spray gun.

Table 1 represents the tensile strength and elongation at the break of the GFRPU according to the weight ratio of the milled glass fiber in the range of 0–7% of polyurea. The tensile strength of the polyurea alone was 22.84 MPa, and the tensile strength of the GFRPU increased with the increase in the content of milled glass fibers. The increase in tensile strength was rather inhibited for the GFRPU with a weight ratio of 7% or more. In this study, the GFRPU reinforcement with a weight ratio of 5% to represent maximum tensile strength was selected to enhance the lateral load-carrying capacity due to the lateral confinement of the column.

The elongation at break gradually decreased with the increase in the content of glass fiber. In addition, the elongation exhibited a tendency to decrease again in the GFRPU of 7% by weight. This was because the higher the proportion of glass fiber, the more the ductility of the polyurea was impaired. The GFRPU with a weight ratio of 5% glass fiber showed an elongation at a break of 362%, ensuring the sufficient ductility of the specimens.

The tensile stress-strain relationship curve of the GFRPU with a weight ratio of 5% is shown in Figure 5 and Figure 6. The curve shown in Figure 5 was divided into an elastic range and a strain-hardening range. It exhibited high ductility until the break. A point inflected slightly from the data within the elastic range (Figure 6) was observed in all specimens. The point commonly occurred at a strain of about 4% for all specimens. It was regarded as the mechanical characteristics of the GFRPU material. The elastic modulus also increased with the increase in the content of glass fiber with the same tendency as the tensile strength (Table 2).

The compressive strength of the concrete was measured with the ϕ100 mm×200 mm cylinders. The cylinders were produced at the same time as the concrete casting, and the strength was measured during the experiment. The concrete used for the footing and column was planned to be 24 MPa, but the experimental results showed an average strength of 27.5 MPa for the footing and 20.4 MPa for the column, respectively.

The longitudinal reinforcing bar used in the specimens was D16, and the shear reinforcing bar was designed as D10. The average yield strength of the D16 was measured at 444.7 MPa, and the tensile strength was 582.3 MPa. The reinforcing bar of D10 exhibited a yield strength of 463.3 MPa and a tensile strength of 567 MPa.

### 3.2. Specimens

Assuming an RC structure with non-seismic reinforcement details, four specimens were designed as SDOF systems based on the fundamental mode. One of them was unreinforced, and the other specimens were laterally strengthened by the GFRPU and dynamically controlled by an LTMD or hybrid control system. The specimens were designed as shown in Figure 7, and bolt holes were predrilled at the bottom of the specimen to fix it on the shaking table. For sufficient anchorage between the footing and the column, the footing reinforcing bars were extended in the direction of the column. Concrete was cast in the footing, the column and the upper slab were reinforced, and the concrete was cast. The mold was removed three days after the concrete was cast, and curing was carried out in the air until the experiment was conducted. Two of the four unreinforced specimens were sprayed with GFRPU; then, we installed the LTMD and hybrid systems.

Figure 2 represents the details of all specimens with and without the control systems. The upper slab weighing one ton corresponded to the overall structure. Figure 8a shows a detailed view of the LTMD rotating at the same angle in the clockwise and counterclockwise directions. The hydraulic dampers and springs in Figure 8b were created in line with the LTMD design requirements and installed. The physical properties were as consistent with the theoretical values as possible.

### 3.3. Loading Instruments and Measurements

The test was performed by a shaking table located at Busan National University Earthquake Disaster Prevention Center (Figure 9). The shaking table specifications are shown in Table 3. The specimens were fixed by fastening them to the shaking table with bolts through the predrilled bolt holes at the bottom of the specimens. The external excitations were applied to the underside of the specimens by the shaking table and controlled by manipulating them remotely from the control room. The side of the specimen on which the LTMD was installed was tightly padded with a steel plate to ensure the close attachment of the LTMD (Figure 10a,b). The deflected state was measured by strain gauges and displacement transducers while loading, as indicated in Figure 10.

## 4. Experimental Results

### 4.1. Natural Frequency of Specimen

Before carrying out this experiment, the first natural frequency was measured to evaluate whether the specimen and the LTMD were tuned. Using the first natural frequency and the corresponding mode shape for a structure, the column and slab of the specimen were designed as an SDOF model. It was not easy to accurately match the theoretical frequency with the actual frequency. The variations in the physical properties of the LTMD can lead to a change in the natural frequency and move beyond the intention to control the vibration. Thus, the frequency was examined by repeated measurements, and the LTMD was calibrated by repeated corrections.

The first natural frequency was estimated by converting the displacement responses in the time domain by the random excitations into the fast Fourier transform (FFT) in the frequency domain (Figure 11). They were measured as 4.5 Hz for the unreinforced specimen, 5.0 Hz for the GFRPU reinforcement specimen, 4.5 Hz for the LTMD control specimen, and 5.25 Hz for the hybrid control specimen, respectively. They were also confirmed by impact hammer tests. In the impact hammer test, the accelerometer was installed at the mid-height of the specimen, where the primary vibration mode was the largest, and the impact was applied with the hammer at the same position. The first natural frequency was the frequency corresponding to the initial maximum amplitude in the frequency response function (FRF) curve, which was the transfer function between the impact force and the corresponding response.

It was observed that the natural frequencies of the GFRPU-reinforced and hybrid control specimens increased due to the slight increase in stiffness by the GFRPU reinforcement. The LTMD control specimen exhibited the same natural frequency as the unreinforced specimen. The LTMD in the hybrid system was manufactured using these properties.

### 4.2. Deflected and Failure Modes

The experiment was performed in the order of the excitation magnitude, as shown in Table 4. Thirteen excitations of one random wave, five sinusoidal waves, and seven actual earthquakes in Korea and El Centro were utilized for this experiment. The dynamic responses and deflected shapes of the specimens were observed while testing. Figure 12 shows the crack patterns for the unreinforced specimen after finishing the experiment.

The dynamic vibration of the specimen less than 6 Hz in the sinusoidal wave experiment was observed with the naked eye. However, the deflected response shape was insignificant, and cracks were rarely confirmed. The first crack occurred in the horizontal direction at the lower center of the column. The cracks gradually propagated to the upper part of the specimen, and the crack width also gradually increased with the increase in the excitation magnitude.

Cracks rarely occurred and developed due to the effect of the vibration control of the LTMD system in the LTMD control specimen. The cracks in the GFRPU reinforcement specimen could not be visually observed because of the GFRPU coating. It was presumed that the crack pattern was similar to that of the unreinforced specimen, but the crack propagation was controlled and delayed due to the tensile strength and ductility of the coated GFRPU.

The cracks were propagated on the entire surface of the column with the increase in the magnitude of the excitation acceleration, but the coating did not peel or fall off. Large vibration responses were observed with the naked eye at a high excitation acceleration. It was estimated that the reinforcing bar yielded, the concrete partially fractured, and the specimen partially failed, although coated by GFRPU.

### 4.3. Dynamic Responses

The vibration responses by a random wave, sinusoidal waves of 4.5 Hz and 6 Hz, and 200% El Centro excitation were compared depending on the proposed techniques. The responses of the sensors indicated in Figure 10c are shown in Figure 13, Figure 14, Figure 15 and Figure 16. In the figures, the numbers in the parentheses after the letter D indicate the sensor positions. D(1), D(2), D(5), and D(8) were located at the same part of the upper slab of the unreinforced, GFRPU, LTMD, and hybrid specimens, respectively. D(3) and D(6) were positioned at the bottom damper of the LTMD and hybrid control specimens, respectively; D(4) and D(7) were at the top damper, respectively.

The dynamic responses were controlled by the reinforcement and control techniques. Figure 13 and Figure 14 exhibit the displacement responses from the 4.5 Hz and 6 Hz sinusoidal wave excitations, respectively. Table 5 summarizes the maximum displacements in the same locations of D(1), D(2), D(5), and D(8). The dynamic responses of the reinforced specimens were reduced from 49.3% to 71.5% in the 4.5 Hz sinusoidal wave excitations. However, they increased from 43.9% to 182.7% in the 6 Hz excitations. It can be shown in the 6 Hz sinusoidal wave that the maximum displacement of the unreinforced specimen was lower than that of the reinforced specimens. This was due to the increase in the displacement as the resonance frequency of the GFRPU reinforcement specimen approached 6 Hz rather than 4.5 Hz. It was also confirmed that the maximum displacement of the LTMD control specimen, which was not reinforced with GFRPU, slightly increased.

Figure 15 and Figure 16 exhibit the displacement responses in the random excitations and 200% El Centro earthquake, respectively. As shown in Table 6, the displacement responses of the GFRPU reinforcement specimen were controlled up to 40.6% in the (+) direction and 23.0% in the (−) direction at random wave excitations and 14.4% and 36.3% in the (+) and (−) directions, respectively, at 200% El Centro excitation. The LTMD control specimen led to the displacement control of 31.0% and 26.9% in the (+) and (−) directions, respectively, at random wave excitations, and 40.8% and 55.2% in the (+) and (−) directions, respectively, at the 200% El Centro excitation. The hybrid control specimen exhibited displacement control of 38.7% and 26.6% in the (+) and (−) directions, respectively, at random wave excitations, and 46.9% and 51.4% in the (+) and (−) directions, respectively, at the 200% El Centro excitation. The displacement responses by the three techniques were controlled efficiently by the enhanced lateral-resisting capacity due to the lateral confinement of the GFRPU reinforcement, the control principle of the TMD, and the control forces constraining the inter-story drift of the LTMD control system.

In this study, the GFRPU reinforcement and the LTMD control techniques were premised on the control of the dynamic responses of the entire structure, not a single specimen. Therefore, the GFRPU reinforcement represented the reinforcement design of all lateral-resisting members, and the LTMD control system was designed to control the lateral responses of the entire structure. Under this assumption, the dynamic responses of the three reinforced specimens were found to be greatly controlled. Since the GFRPU technique strengthened individual members, it could not control the displacement responses of the entire structure as much as the vibration control by the LTMD control system. Considering that it was the vibration control of the entire structure rather than a single member, the displacement responses were more controlled in the order of the GFRPU reinforcement, the LTMD control, and the hybrid control. Larger responses in the reinforced specimens than in the unreinforced specimen within some time ranges were observed because the displacement responses were partially governed by the resonance frequency.

The efficiency of the control techniques was evaluated by comparing the displacement responses of the upper slab. Figure 17 compares the displacement responses depending on the three techniques at seismic excitations and random waves. The dynamic responses were greatly alleviated by the control systems presented in this study. The hybrid control was concluded to be the most reliable seismic reinforcement technique.

### 4.4. Control of Dynamic Responses

The lateral-resisting capacity due to the lateral confinement of the column by the GFRPU was enhanced. The GFRPU reinforcement exhibited more improved lateral resistance than the unreinforced specimen. The lateral-resisting capacity can be explained as the control force, which is the enhancement in the shear-resisting capacity to control the displacement responses. Assuming a constant mass for all specimens, the control force was analyzed as the variation in the accelerations before and after the reinforcement. The strengthening and control techniques were evaluated by the control forces to reduce the displacement responses at the upper slab of the specimen.

Figure 18 represents the acceleration responses under 100% El Centro excitations. The difference in acceleration between the unreinforced and reinforced specimens can be explained by the action of the control force. It was presumed from the plot that the control forces acted strongly on the specimens from the high reduction in the accelerations. Additional control forces acted on the lower part of the LTMD control specimen other than its upper part, which greatly reduced the vibration responses compared to the GFRPU reinforcement specimen. It can be observed that the acceleration responses of the upper slab were significantly reduced in the order of the GFRPU, LTMD, and hybrid reinforcements. The shear-resisting capacity or control force was increased in this order.

## 5. Conclusions

This research was performed to investigate the strengthening techniques of RC structures with non-seismic reinforcement details. The strengthening techniques were developed by the composite elastomer for enhancing the lateral stiffness, ductility, and dissipated energy of the members and/or a passive control system. This study verified the effectiveness of the GFRPU reinforcement, LTMD control, and hybrid control techniques by comparing the displacement control capacity and control force with the unreinforced specimen. The GFRPU reinforcement enhanced the lateral resistance capacity and ductility of the specimens. Applying the vibration control concept of TMD and importing the constraint characteristics to control the inter-story drift, the LTMD reinforcement technique was proposed, and its efficiency was illustrated. The hybrid control system improved the lateral resistance capacity as well as the ductility of the specimen to prevent abrupt failure with the ductility of the GFRPU. The hybrid control system should be designed to control the vibration responses of the entire structure by the LTMD and to secure the lateral rigidity and ductility of the local members by the GFRPU. It was displayed that the displacement responses of the reinforced specimens were controlled up to 46.9% and 55.2% in the (+) and (−) directions under the ground excitations. It was observed that the control capacity of the dynamic responses was increased in the order of the GFRPU, LTMD, and hybrid reinforcements.

## 6. Recommendations

The GFRPU reinforcement is utilized to locally strengthen the degraded members. The LTMD system is effective for the global vibration control of a structure. Contrary to the original purpose, the GFRPU reinforcement was considered a global reinforcement rather than a local reinforcement because an entire structure was transformed into an SDOF system. Therefore, the hybrid reinforcement technique will be analyzed, designed, and studied separately for controlling dynamic responses.

## Figures and Tables

**Figure 1 polymers-14-04979-f001:**
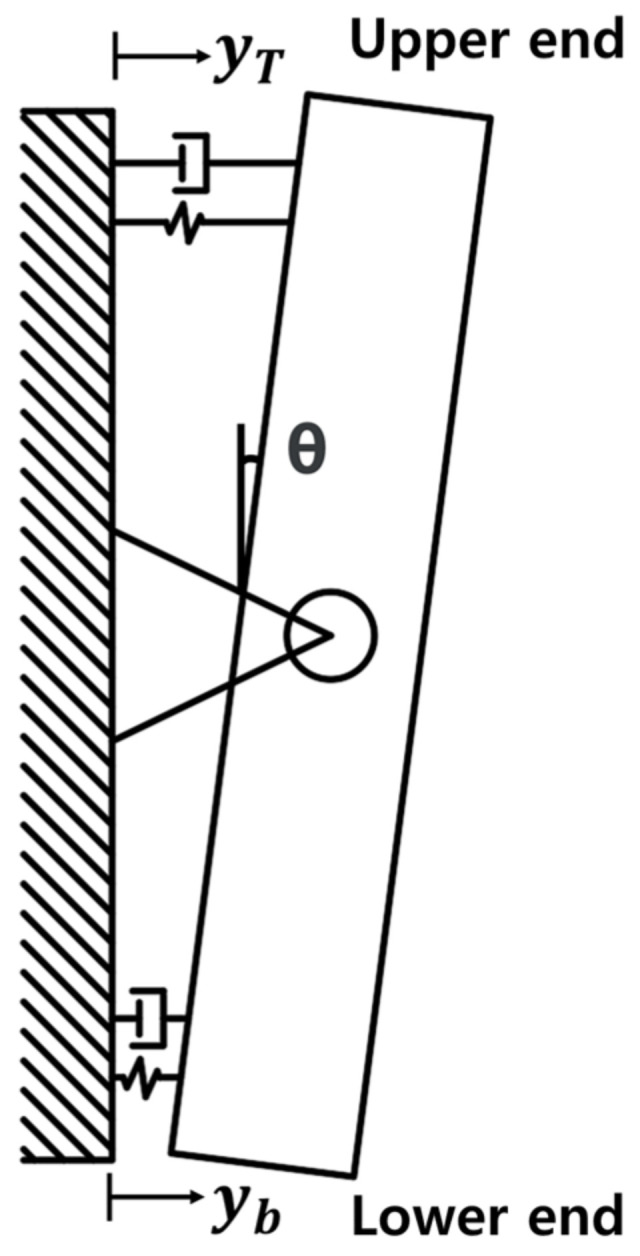
Mechanical behavior of LTMD.

**Figure 2 polymers-14-04979-f002:**
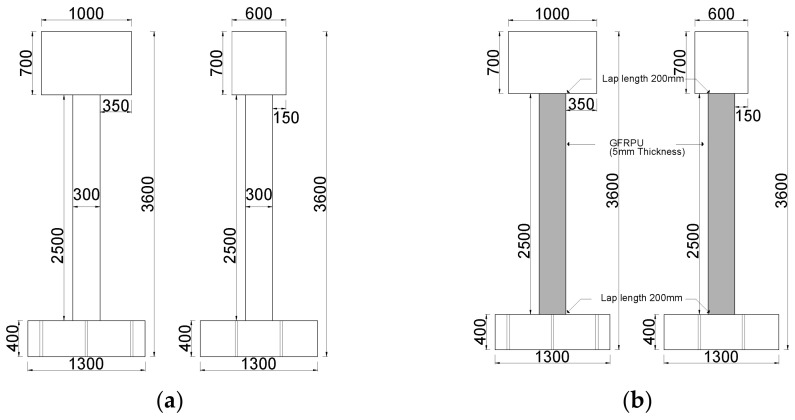
Four specimens: (**a**) unreinforced specimen, (**b**) GFRPU-reinforced specimen, (**c**) LTMD control specimen, and (**d**) hybrid control specimen (unit: mm).

**Figure 3 polymers-14-04979-f003:**
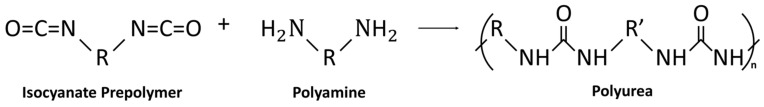
Polyurea reaction scheme.

**Figure 4 polymers-14-04979-f004:**
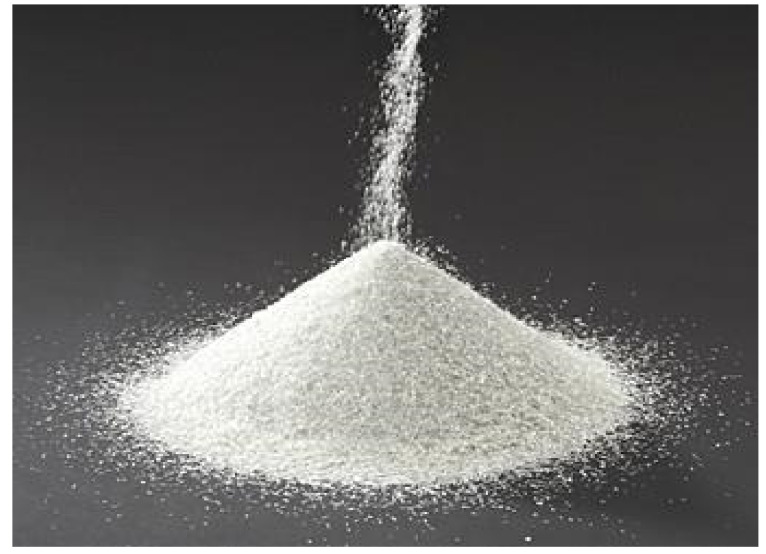
Milled glass fiber.

**Figure 5 polymers-14-04979-f005:**
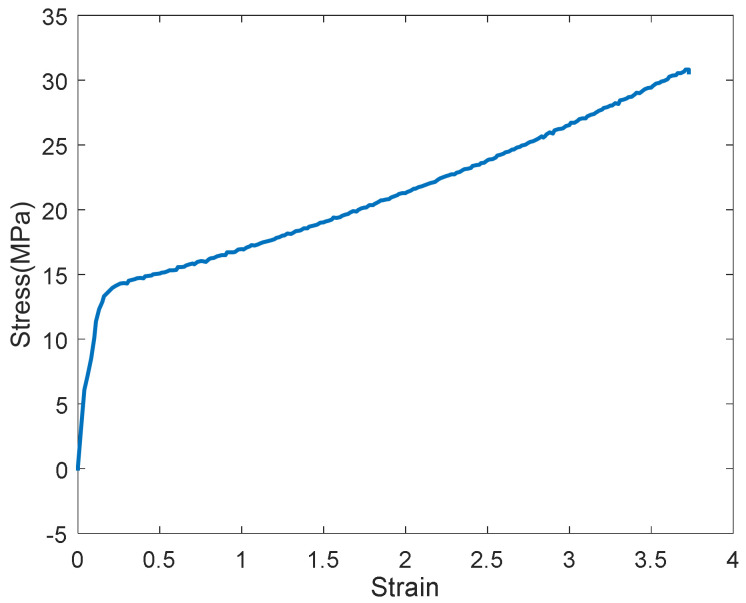
Tensile stress-strain curve of 5% GFRPU.

**Figure 6 polymers-14-04979-f006:**
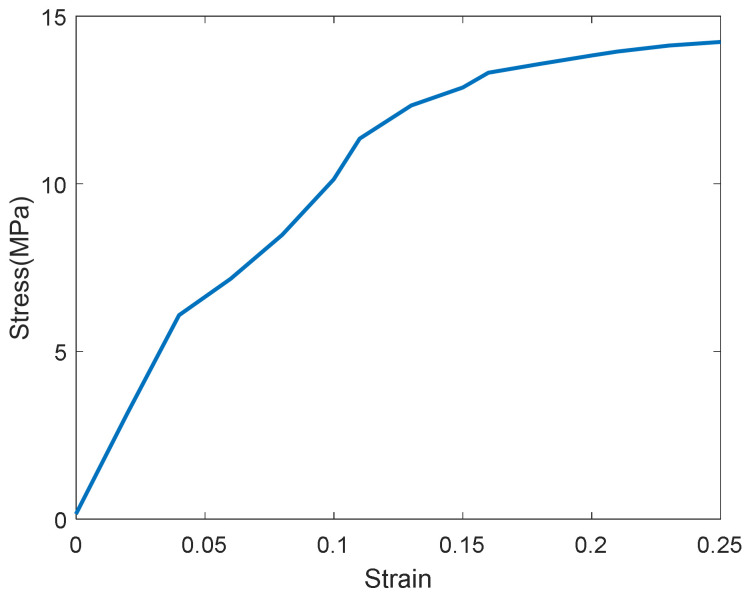
Tensile stress-strain curve of 5% GFRPU in the elastic range.

**Figure 7 polymers-14-04979-f007:**
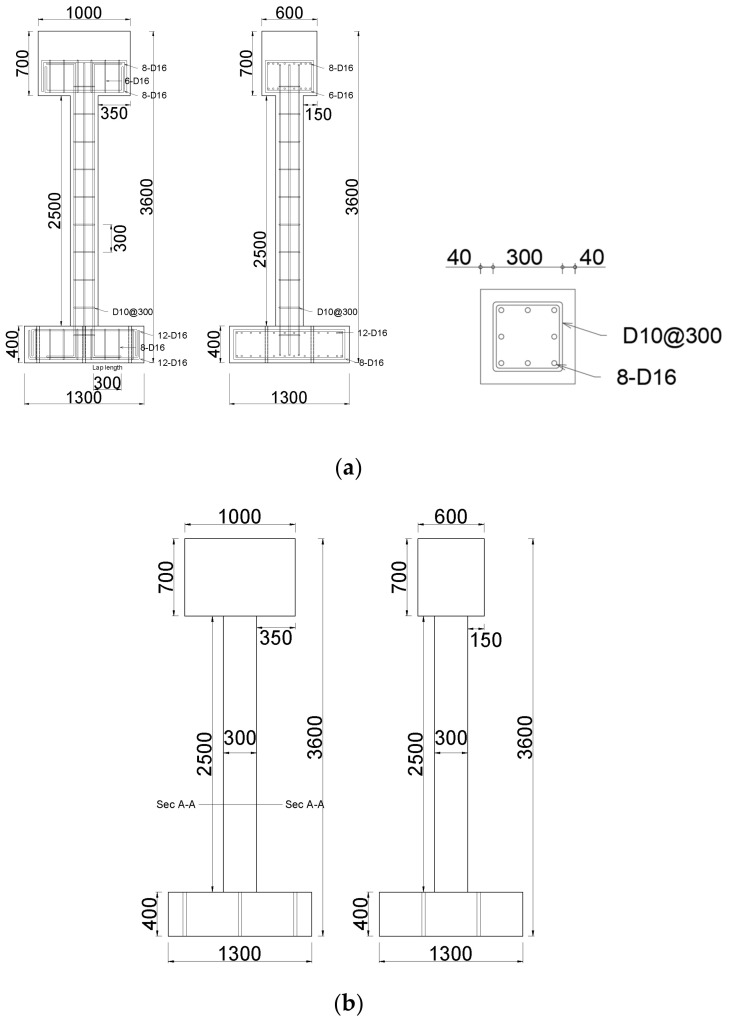
Details of the specimen reinforcement: (**a**) reinforcement details of the specimen and the (**b**) details of the specimen holes (unit: mm).

**Figure 8 polymers-14-04979-f008:**
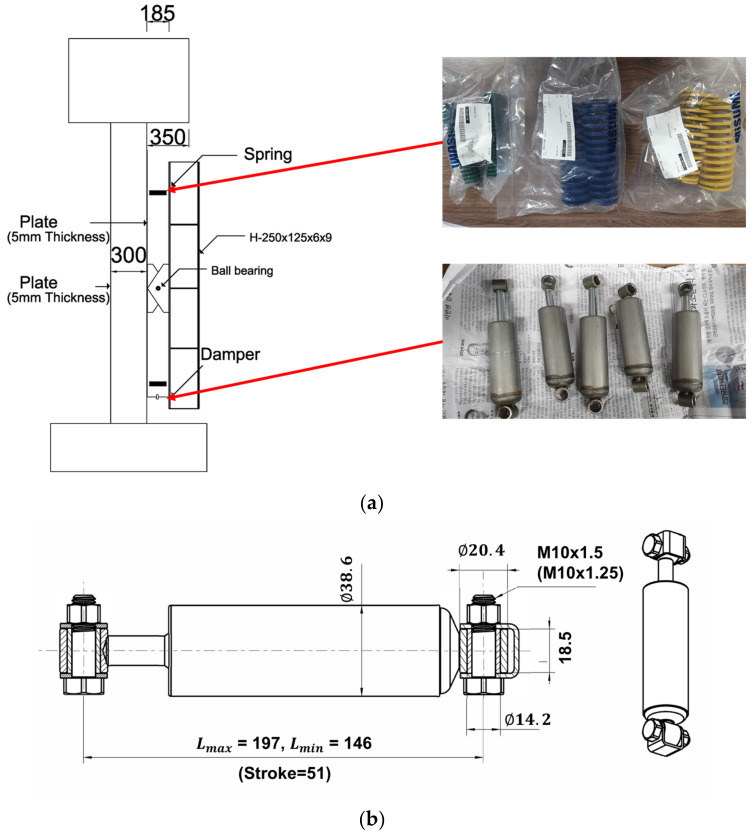
The LTMD system: (**a**) details, (**b**) hydraulic damper (unit: mm).

**Figure 9 polymers-14-04979-f009:**
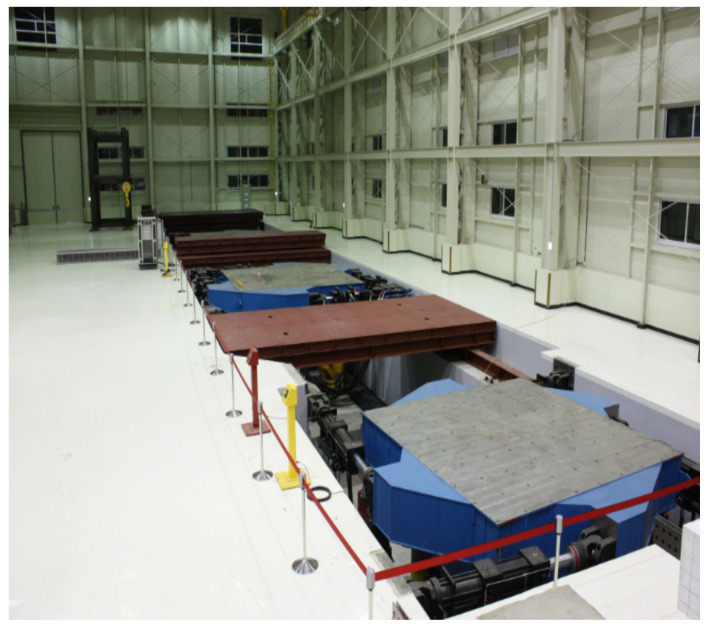
A shaking table.

**Figure 10 polymers-14-04979-f010:**
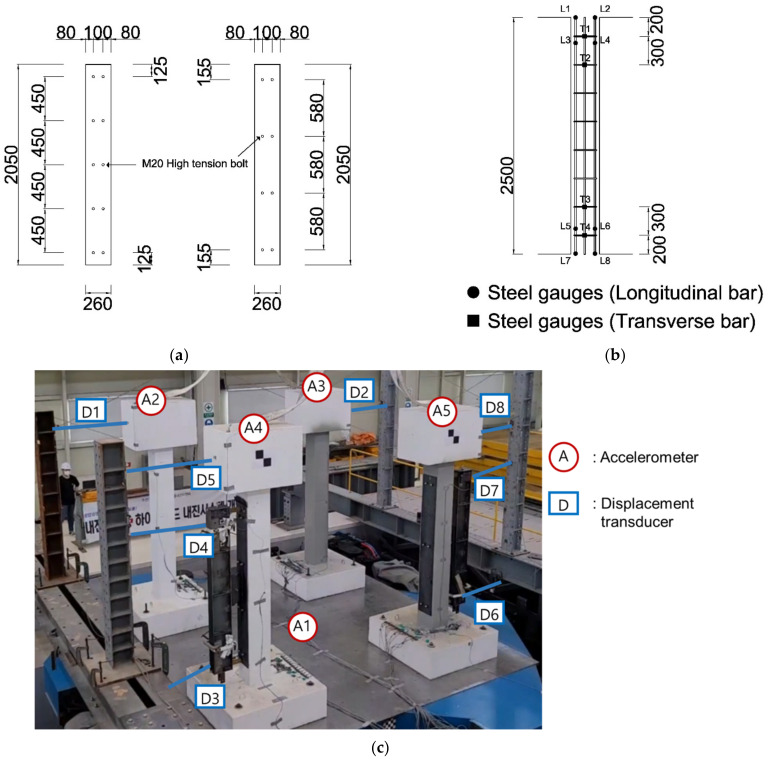
Locations of the measurement sensors: (**a**) attachment of the steel plate at the column section, (**b**) locations of the strain gauges, and the (**c**) locations of the accelerometers and displacement transducers (unit: mm).

**Figure 11 polymers-14-04979-f011:**
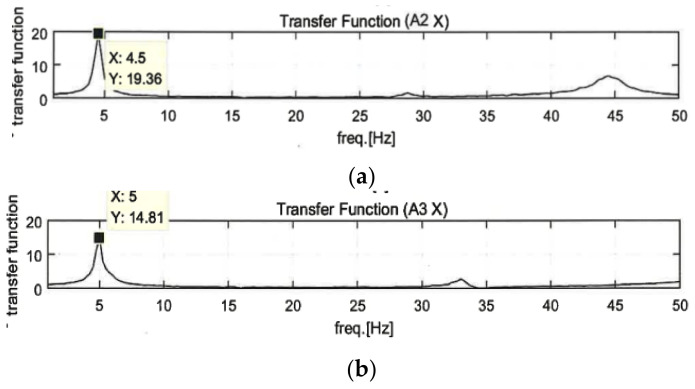
FFT of the displacement responses: (**a**) unreinforced specimen, (**b**) GFRPU reinforcement specimen, (**c**) LTMD control specimen, and the (**d**) hybrid control specimen.

**Figure 12 polymers-14-04979-f012:**
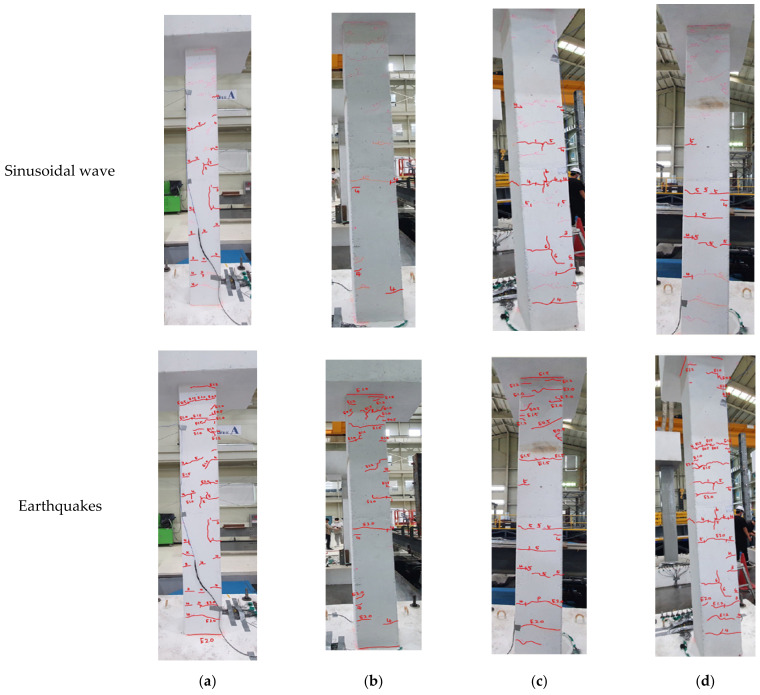
Crack patterns of the unreinforced specimen: (**a**) front view, (**b**) back view, (**c**) right-side view, (**d**) left-side view.

**Figure 13 polymers-14-04979-f013:**
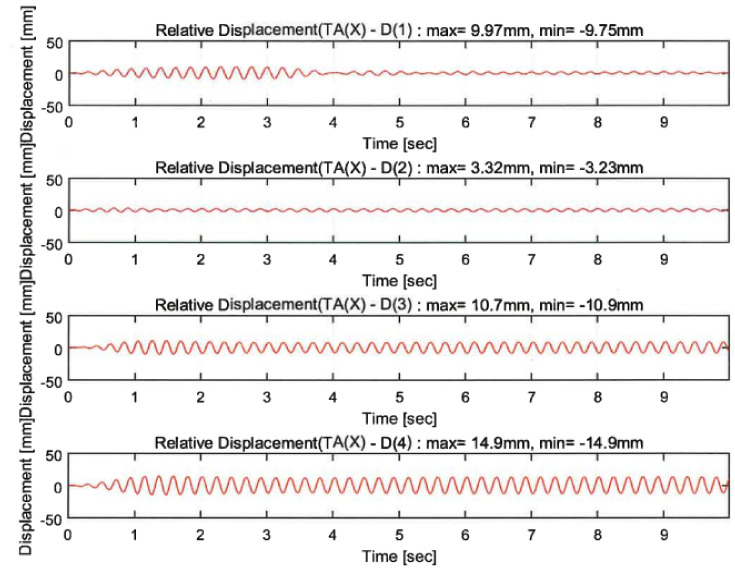
Dynamic responses under a sinusoidal wave of 4.5 Hz.

**Figure 14 polymers-14-04979-f014:**
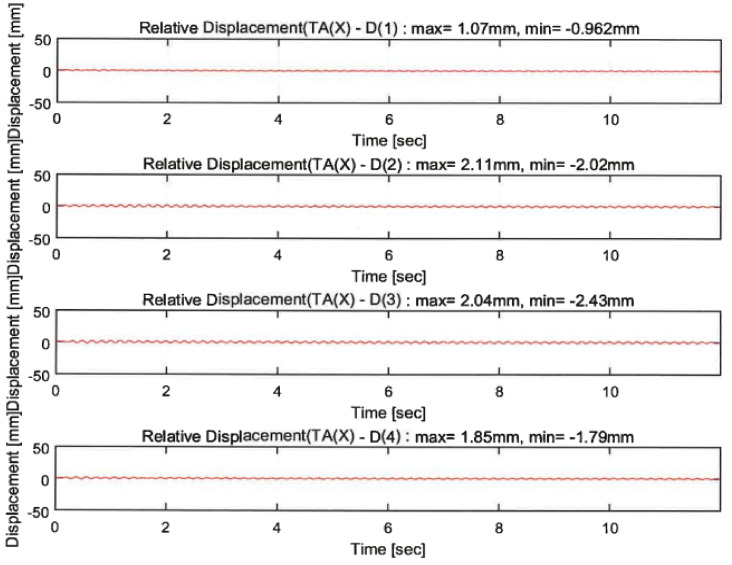
Dynamic responses under a sinusoidal wave of 6.0 Hz.

**Figure 15 polymers-14-04979-f015:**
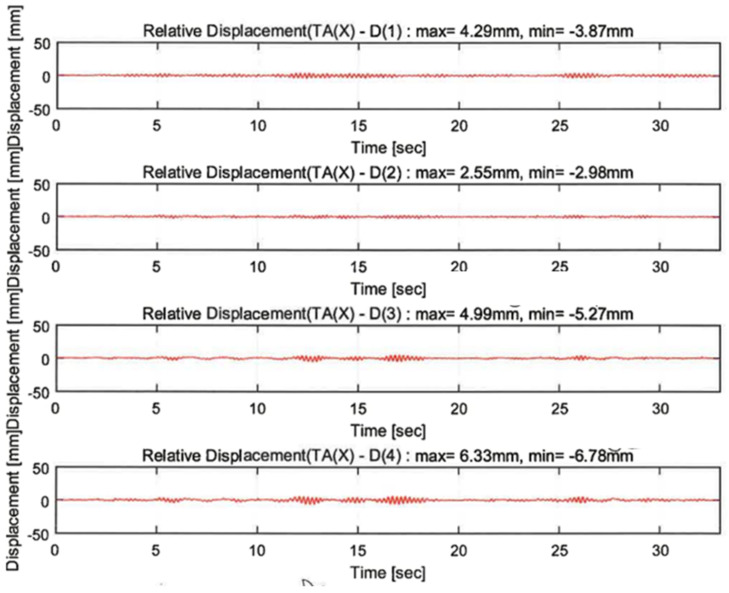
Dynamic responses under random excitation.

**Figure 16 polymers-14-04979-f016:**
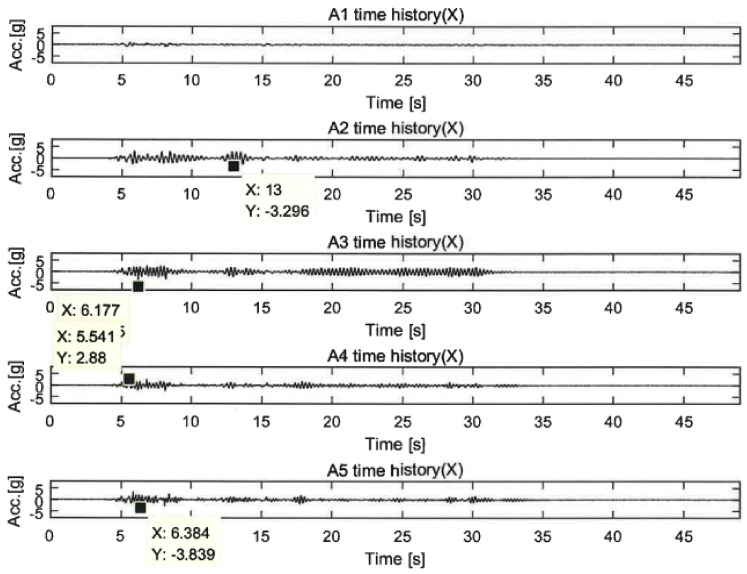
Dynamic responses under 200% El Centro earthquake.

**Figure 17 polymers-14-04979-f017:**
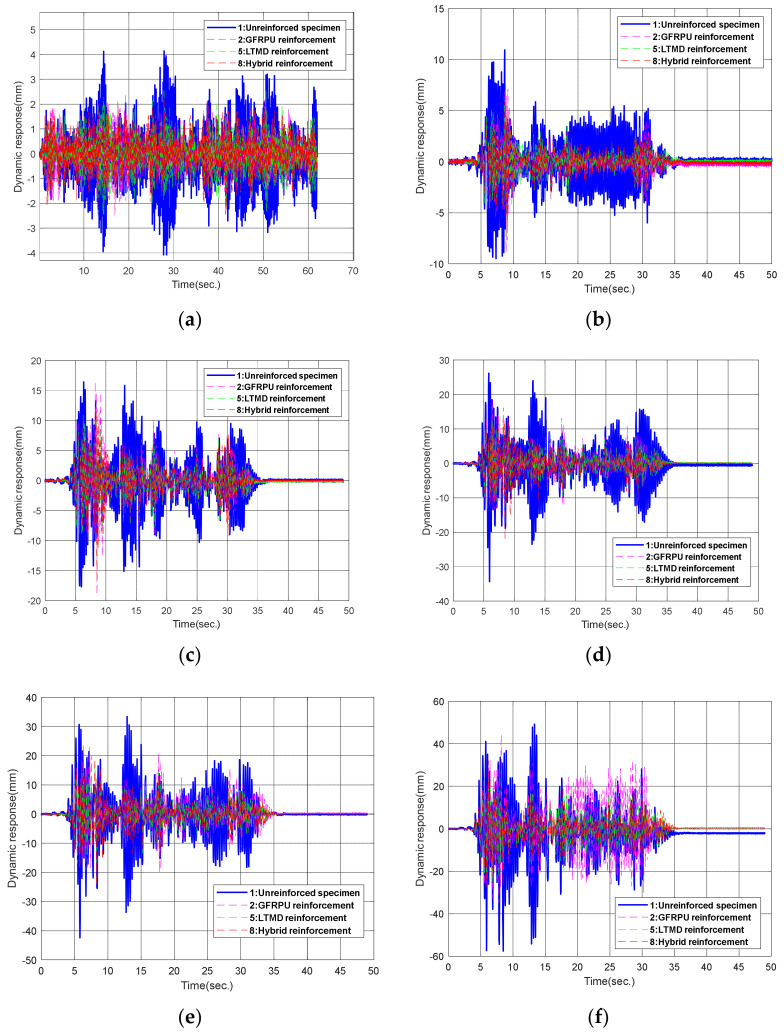
Displacement response curves: (**a**) random, (**b**) 50% El Centro, (**c**) 100% El Centro, (**d**) 120% ElCentrol, (**e**) 150% ElCentrol, (**f**) 200% El Centro.

**Figure 18 polymers-14-04979-f018:**
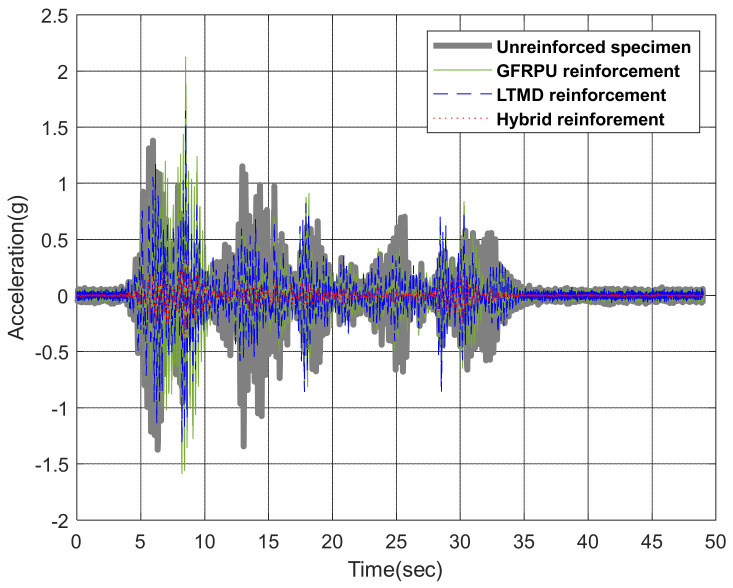
Variation in the acceleration under the 100% El Centro earthquake.

**Table 1 polymers-14-04979-t001:** Physical properties of GFRPU according to glass fiber content.

	0%	1%	3%	5%	7%
Average tensile strength (MPa)	22.84	29.33	29.40	30.96	30.25
Elongation at break (%)	418	396	355	362	352

**Table 2 polymers-14-04979-t002:** Elastic modulus of GFRPU according to glass fiber content.

	0%	1%	3%	5%	7%
Elastic modulus (MPa)	99.89	119.83	134.95	143.90	143.20

**Table 3 polymers-14-04979-t003:** Specification of shaking table.

Max. Loading	60 ton
Table size	5.0 m × 5.0 m
Control Axes	3 DOF (Translational 2 axes, rotational 1 axis)
Max. Displacement	X-Axis = ± 300 mm, Y-Axis = ± 200 mm
Max. Velocity	*H* = 1000 m/s
Max. Acceleration at Bare Table	±3.0 g
Frequency Range	(0.1~60) Hz
Excitation Mechanism	Electro-hydraulic Servo, 3 Variable Control
Control Software	MTS 469D, STEX Pro
Feedback Data Acquisition	51 Channels (Sampling rate = 512 Hz)

**Table 4 polymers-14-04979-t004:** Thirteen excitations.

Number of Specimens	Excitations
1	Random wave
2~6	Sinusoidal wave (3 Hz, 4.5 Hz, 6 Hz, 7.5 Hz, 9 Hz)
7~13	Earthquake (Kyungju and Pohang in Korea, El Centro (50%, 100%, 120%, 150%, 200%))

The percentage in the parentheses of El Centro represents the amplification of the El Centro earthquake.

**Table 5 polymers-14-04979-t005:** Summary of dynamic responses under sinusoidal waves.

	D(1) Unreinforced		D(2) GFRPU		D(5) LTMD		D(8) Hybrid	
	(+)	(−)	(+)	(−)	(+)	(−)	(+)	(−)
4.5 Hz sinusoidal wave (mm/%)	9.97	−9.75	3.32/ 66.7	−3.23/ 66.9	4.67/ 53.2	−4.94/ 49.3	2.84/ 71.5	−2.78/ 71.5
6 Hz Sinusoidal wave (mm/%)	1.07	−0.962	2.11/ −97.2	−2.02/ −110.0	1.54/ −43.9	−1.39/ −44.5	2.78/ −160	−2.72/ −182.7

**Table 6 polymers-14-04979-t006:** Summary of dynamic responses under random excitation and 200% El Centro.

	D(1) Unreinforced		D(2) GFRPU		D(5) LTMD		D(8) Hybrid	
	(+)	(−)	(+)	(−)	(+)	(−)	(+)	(−)
Random excitation (mm/%)	4.29	−3.87	2.55/ 40.6	−2.98/ 23.0	2.96/ 31.0	−2.83/ 26.9	2.63/ 38.7	−2.84/ 26.6
200% El Centro (mm/%)	50.8	−56.2	43.5/ 14.4	−35.8/ 36.3	30.1/ 40.8	−25/ 55.2	27/ 46.9	−27.3/ 51.4

## Data Availability

The data used to support the findings of this study are included within the article.

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
