# Peer review of "Comparison of Dynamic Vibration Control Techniques by the GFRPU and/or LTMD System"

_polymers, 2022, doi:10.3390/polym14224979_

Round 1

Reviewer 1 Report

Full title: Dynamic vibration control by a Hybrid system of GFRPU and 1 LTMD

Ms. Ref. No.: polymers-2006349

The authors have conducted experimental tests to investigate the seismic performance enhancement of structures using (1) glass fiber-reinforced polyurea (GFRPU) reinforcement, (2) a lever-typed tuned mass damper (LTMD) system, and (3) a hybrid system of GFRPU and LTMD reinforcements.

The subject addressed is within the scope of the journal. There are simply many grammar mistakes. The manuscript needs language, grammar, and syntactic editing. The English language usage should be checked by a fluent English speaker. It is understood that for the non-English speaker it is ok, but it still needs to be properly checked and proofread it. However, the manuscript, in its present form, contains several weaknesses. Appropriate revisions to the following points should be undertaken in order to justify the recommendation for publication.

(1) The authors should explain the “load-carrying mechanism” of the three adopted techniques. How the load is being transferred? how it is carried by them?? And more importantly how the structure respond to the applied technique??

 (2) The authors should discuss the failure modes obtained or captured in the study.

 (3) Damage propagation mechanism should be explained in brief with experimentally captured figures.

 (4) The title should be rephrased because is not suitable.

 (5) The introduction section is too weak and no deep discussion could be found related to current design codal provisions on such techniques. In the review of literature, the latest development should be highlighted rather than piling up the paper.

Recommended references:

:- "Experimental and Numerical Investigations on Performance of Reinforced Concrete Slabs under Explosive-induced Air-blast Loading: A state-of-the-art review", Structures, Elsevier, 31: 428-461, https://doi.org/10.1016/j.istruc.2021.01.102.

 More references need to be added.

 (6) It is advised to include recent literature which discusses the behavior of buildings under quasi-cyclic loading.

 (7) What is the main contribution of this paper? What motives for that? What gap in this field has been covered? Which part of the mentioned instruction has been progressed? According to what criteria?

 (8) Comparison of the measure displacements with the permissible limits given in different design standards/manuals/codes should be incorporated.

 (9) Some assumptions are stated in various sections. Justifications should be provided on these assumptions. Evaluation on how they will affect the results should be made.

  (10) Conclusion section is weak. There are no quantitative points.

 (11) Lastly, must add recommendations that derive from your analysis...do you recommend reviewing the Regulation/Direction provided by the government for the design of earthquake-resistant structures? Is any change needed??

The main concern about the manuscript is its contribution to knowledge. It is expected that a critical gap analysis will be done in the introduction section to justify the necessity of doing this piece of research. It is partially done by the authors but it should consider all aspects of the problem including assumptions, limitations and constraints, pros and cons, and relative merits to the other publicly available studies and proposals.

 Recommendation:

Double Major Revision: If the authors of the manuscript are able to address all the aforementioned issues and make the necessary amendment accordingly, the article may possibly be published in “Polymers” Journal of MDPI.

Reviewer 2 Report

After a comprehensive review of the manuscript "Dynamic vibration control by a Hybrid system of GFRPU and LTMD", I am recommending it for publication after processing major revisions to the comments:

1.      Please modify the abstract that cover all aspects.

2.      Reduce the keywords to six

3.      Add more details about glass fiber-reinforced polyurea (GFRPU) reinforcement and lever-typed tuned mass damper (LTMD) in the introduction.

4.      In order to make your research more attractive to readers and researchers, I suggest that you include a new section entitled "Research significance".

5.      Due to the large number of abbreviations throughout the manuscript, it is suggested to include a table of abbreviations

6.      There is no citation for Figure 2 in the text

7.      Figure 3 needs a citation

8.      The manuscript needs to address some technical issues such as capitalization within sentences, commas, superscripts for units and chemical symbols.

9.      A comparison with previous study is necessary.

10.  Standardization of writing hyphenated phrases throughout the manuscript, whether with or without hyphens. Apply this note to all words that contain a hyphens.

11.  It is preferable to add a section "Recommendations" to indicate the authors' recommendations for future researchers.

12.  The manuscript needs a minor linguistic revision.

13.  Revise the conclusions based on previous comments and new revisions.

14.  Add references from recent years.

Round 2

Reviewer 1 Report

The authors have made the corrections as suggested by the reviewers, therefore the paper can be accepted for publication in this journal. 

Reviewer 2 Report

Now, I recommend the revised manuscript for acceptance for publication.